# Double Transverse Foramina—An Anatomical Basis for Possible Vertebrobasilar Insufficiency Risk and Vertebral Artery Injury

**DOI:** 10.3390/diagnostics13193029

**Published:** 2023-09-23

**Authors:** Nilgün Tuncel Çini, Shahed Nalla, Federico Mata-Escolano, Esther Blanco-Perez, Juan José Valenzuela-Fuenzalida, Mathias Orellana-Donoso, Juan A. Sanchis-Gimeno

**Affiliations:** 1Department of Anatomy, Faculty of Medicine, Bilecik Seyh Edebali University, Bilecik 11100, Türkiye; nilgun.cini@bilecik.edu.tr; 2Department of Human Anatomy and Physiology, Faculty of Health Sciences, University of Johannesburg, Johannesburg 2092, South Africa; juan.sanchis@uv.es; 3GIAVAL Research Group, Department of Anatomy and Human Embryology, Faculty of Medicine, University of Valencia, 46001 Valencia, Spain; 4Department of Radiology, ERESA CT and MRI Unit, 46015 Valencia, Spain; fmatae@ascires.com; 5Department of Radiology, University Hospital de la Ribera, 46600 Alzira, Spain; 6Department of Morphology, Faculty of Medicine, Universidad Andres Bello, Santiago 8370134, Chile; juan.valenzuela@unab.cl (J.J.V.-F.); mathias.orellana@unab.cl (M.O.-D.); 7Department of Morphology and Function, Faculty of Health and Social Sciences, Universidad de las Américas, Santiago 7500975, Chile

**Keywords:** anatomic variations, cervical vertebrae, computed tomography, double transverse foramen, spine, vertebral artery, vertebrobasilar insufficiency

## Abstract

Cervical vertebrae may exhibit the anomalous presence of a double transverse foramen (DTF) that may impact the anatomy of related structures that traverse the cervical region, such as the vertebral artery (VA). This retrospective anatomical study utilized CT angiography cervical scans to examine the prevalence of DTF, VA, and TF areas. The subjects were separated into two groups: normal TF (NTF group; 26 males and 21 females) and double TF (DTF group; 21 males and 24 females). The males presented significantly higher TF area values (30.31 ± 4.52 mm^2^) than the females (27.48 ± 1.69 mm^2^) in the NTF group (*p* = 0.006). The sex differences disappeared when a DTF was present (*p* = 0.662). There were no differences in the VA area values between the sexes in both the NTF and DTF groups (*p* = 0.184). No significant differences in the VA area values between males of the NTF and DTF groups (*p* = 0.485) were noted. The DTF subjects presented an increased VA/TF area ratio than the NTF subjects (*p* < 0.001). This study showed that DTF presence reduced the TF area. In contrast, the VA area did not change despite the decreasing TF area. This might be an anatomical risk for transient vertebrobasilar insufficiency in subjects with DTF, especially in females, because VA space in the TF is less in DTF subjects than in NTF subjects. This may lead to easy VA compression in DTF subjects following neck trauma.

## 1. Introduction

The most important specific anatomical feature that distinguishes the cervical vertebrae from other vertebrae of the vertebral column is the transverse foramen (TF) contained within the transverse processes, which are bony projections that extend laterally from the vertebra, serving as attachment sites for muscles, ligaments, and other structures [1]. The TF is formed by the union of the transverse process vertebral element posteriorly and the remains of the cervical rib vertebral element anteriorly. The cervical rib is a rudimentary anomalous rib that may present in some individuals. It is usually absent in the first six cervical vertebrae but may be present in the seventh vertebra.

The TF provides a protective cervical passage for the vertebral artery (VA), the vertebral vein, and the fibers of a sympathetic nerve plexus [2]. The course of the VA originating from the first part of the subclavian artery en route to the brain is divided anatomically into four parts (Figure 1). The part starting from the subclavian artery to the TF on C6 is defined as V1 (or a prevertebral part), the part that runs in the TF between C6 and C1 (atlas) is defined as V2 (or the foraminal part), and the part between the sulcus and foramen magnum on the atlas vertebra is defined as V3 (or the suboccipital part). Finally, the part that traverses within the skull is V4 (or the intradural part). VA injuries are one of the lead causes of many pathologies and diseases, such as cerebral ischemia, pseudoaneurysm, thrombosis, and embolism [3]. While traumatic vertebral artery injuries are seen in the V2 segment in adults [4], it is stated that they occur in the upper V2 and V3 segments in infants and children [5].

The standardization of the human body has resulted in the establishment of how organs and systems should appear (i.e., normality), with such descriptions being idealized images of the “perfect body” [6]. But as suggested by Zytkowski et al. [6], anatomical variations are not “unnatural”, and an awareness of them is essential for “successful” medical practice. Thus, anatomical variability may be taught in medical schools and kept in mind by clinicians.

In this context, the normal C6 cervical vertebrae may present only one FT, but the variations that are observed to be related to the TF and their anatomical details are essential for physicians and radiologists to interpret medical images [7]. Variations in the size and shape of the TF are common. In some people, the foramen may be larger or smaller than normal. It may also be blocked or narrowed by bone or other tissue. These variations can affect the blood supply to the brain and spinal cord. Variations are often seen as hypoplastic TF, a duplication of TF, or shape and size variability exhibited by the TF. Abnormalities and variations of the TF can sometimes compress the vertebral artery or vein, leading to pain, numbness, and weakness in the neck, arms, and hands. In severe cases, it can even cause a stroke. In complex surgical procedures, it is crucial to understand the TF variations and, thus, potentially affected VA variations to avoid potential complications [8]. Secondary vascular compression can be seen due to possible lesions, traumas, or general fractures or luxations in the VA due to the TF [9].

One of the most common variations is the bisection of the TF, or the double transverse foramen (DTF), which is defined as an accessory foramen and is reported to be most common on the C6 [10,11]. Variations in the course and development of the VA are likely to be associated with variations in the TF [2]. In this context, computed tomography angiography (CTA) allows for the analysis of the DTF and the VA in vivo (Figure 2, Figure 3 and Figure 4).

Vertebrobasilar insufficiency (VBI) is indicated by insufficient blood flow in the posterior circulation path of the brain, provided by the basilar artery that is formed by the union of the right and left Vas, and is often used to describe the transient ischemic attacks in the vertebrobasilar territory [12]. Factors that exacerbate atherosclerosis, such as smoking, hypertension, age, gender, family history, and genetics, predispose patients to VBI, in which case it is possible to say that patients with a history of coronary artery disease or peripheral artery disease are at a higher risk [13]. Most strokes and transient ischemic attacks occur in the vertebrobasilar territory. It is stated that ethnicity is a factor as much as gender in intracranial atherosclerotic disease [14], and is estimated to cause 10–15% of ischemic strokes in Western countries [15]. This rate rises to 54% in Asian populations. It is now recognized as the leading cause of death in the Chinese population [16]. These differences are significant in creating safe areas in surgical screw fixation. For these reasons, morphometric and anatomical examinations of TF variations according to age, gender, and laterality are essential for understanding various pathologies and developing treatment methods [17].

In our research, we hypothesize that the presence of a DTF reduces the TF area, while it does not affect the VA area. The positive confirmation of a decreased TF area in subjects with an unchanged VA area may lead to a reduced VA/TF area ratio in DTF subjects. We hypothesized a reduced VA/TF area ratio in DTF subjects as a potential anatomical risk that could lead to transient VBI and transient ischemic strokes under some circumstances, like whiplash or neck trauma, due to a higher risk of VA compression as the VA has less space inside the TF in DTF subjects, and that it may lead to an easy compression of the VA in DTF subjects following whiplash or neck trauma.

## 2. Materials and Methods

The study was carried out by forming two groups. The first group included individuals with NTF (NTF group or normal anatomy group), while the second group included individuals with DTF (DTF group). This anatomical-based, retrospectively designed single-center study was approved by the Ethics Committee in Human Research of the University of Valencia (ref. H1414410627187). The procedures were performed in accordance with the World Medical Association’s Declaration of Helsinki.

Inclusion criteria were Spanish individuals aged 18 years and over in whom TF was seen, and VA progressed starting from C6. The reason for performing the CTA scans analyzed in this study was suspected internal carotid artery diseases. Two different radiologists reviewed the CT scans. The CT images were used for the study only when both radiologists concurred in their diagnosis.

The exclusion criteria were the following: studies in which the complete cervical spine was not included, metal artefact imaging, previous cervical spine surgery, trauma or vertebral fractures, tumor history or cervical spine infections, severe rheumatic disease, myelopathies and congenital cervical malformations, and any pathological findings and/or malformations of VA. Based on our inclusion and exclusion criteria, 92 (74.2%) CTA scan images of the cervical spine were collected from a total sample of 124 Spanish subjects (100%). The CTAs were performed at the ERESA CT Unit of Valencia, Spain, and all subjects signed a written consent form allowing for these data to be used for scientific purposes.

As a result, 92 subjects (100%) were included in the NTF or the DTF group. A total of 47 (51.1%) individuals, 26 males (55.3%) and 21 females (44.7%), were included in the NTF group (the mean age of males was 36.38 ± 7.97 years, and the mean age of females was 36.72 ± 8.97 years). In comparison, a total of 45 (48.9%) individuals, 21 males (46.7%) and 24 females (53.3%), were included in the DTF group (the mean age was 33.33 ± 6.25 years and 37.00 ± 7.28 years for males and females, respectively).

Imaging studies for this project utilized the GE LightSpeed VCT 64 Slice CT system (General Electric, Milwaukee, WI, USA). This system offers an axial field of view spanning 350–400 mm with a transaxial slice thickness of 0.5 mm. To measure TF (Total Fat) and VA (Vascular Area), Autobone and VessellQ Xpress software (General Electric, Milwaukee, WI, USA. https://www.gehealthcare.es/products/advanced-visualization/all-applications/autobone-vesseliq-xpress (accessed on 20 August 2023)) were utilized. A low-dose CT scan was conducted with each patient lying in the supine position, encompassing the aortic arch to the orbitomeatal baseline. The scan parameters included 120 Kvp, 300 mAs, a 0.5 s rotation time, and a pitch of 0.64. An 18- to 20-gauge angiocath was inserted into the antecubital vein for each patient. A contrast medium, nonionic iodinated iopamidol 300, ranging from 70 to 120 mL (2 mL per kg of body weight), was administered via an automatic injector, followed by 30 mL of saline solution at a rate of 5 mL/s. In order to accurately visualize the vertebral arteries, a bolus-tracking method was employed. The region of interest was placed at the aortic arch, with a threshold set at 100–120 Hounsfield units (HU). When this threshold was exceeded, helical scanning was automatically initiated.

The area of the TF (mm^2^) and the VA (mm^2^) were measured for both the right and left sides of the vertebra. The TF measured in the DTF group was the principal TF (the TF where the VA was located inside it), and not the accessory TF. 

Statistical analysis was conducted using the program SPSS 25.0 (IBM). Descriptive statistics were given as mean ± standard deviation (S.D.) for variables. The Kolmogorov–Smirnov test was utilized to determine the normal distribution of the data. Student’s *t*-test and Mann–Whitney U test were used for parametric and non-parametric variables to compare sides and determine the gender differences. A *p*-value of <0.05 was considered significant.

## 3. Results

The descriptive statistical values for males and females and the side differences in the two groups are shown in Table 1 and Table 2. An analysis of the results presented in Table 1 and Table 2 revealed no significant differences between the right vs. left TF and VA area values found in the males and females of both the NTF and DTF groups. 

The males presented significantly higher TF area values than the females in the NTF group, but the differences between genders disappeared when a DTF was present. In addition, no differences in the VA area values between the sexes in both the NTF and DTF groups were found (Table 3).

The males and females of the NTF group presented significantly higher TF area values than those of the DTF group (Table 3). There were no significant differences in the VA area values between the males of the NTF and DTF groups. In contrast, the females of the DTF group presented significantly higher VA area values than the females of the NTF group (Table 3).

The analysis of the ratio of VA/TF is presented for both groups in Table 4. The ratio of VA/TF for all of the DTF subjects was 0.48 ± 0.09, while it was 0.66 ± 0.07 for all of the NTF subjects (*p* < 0.001). There was sexual dimorphism in the VA/TF ratio of the normal anatomy subjects (NTF group), with the females having higher values than the males. Still, this sexual dimorphism disappeared when a DTF was present (Table 4). In addition, the ratio VA/TF area was higher in both the males and females of the DTF group compared to the males and females of the NTF group (Table 4).

## 4. Discussion

We showed the sexual dimorphism of TF and the VA domains related to this study. Since it is not possible to study the area of the VA in dry bones, we tried to show this dimorphism in vivo with CTA images [18,19] because CTA allows for the production of detailed images of both the bony structures (the TF) and the vessels (the VA). Thus, we used CTA because it allowed us to analyze the in vivo relationship between the TF and the VA and to measure the TF and VA areas and because it was proposed that using CTA is a good solution for the in vivo study of the DTF and the VA [19].

In this study, while there was a difference between the males and females in the presence of a single TF (NTF group), this difference disappeared in the presence of a second foramen (DTF group). In addition, although the TF area was higher in males, there was no difference between the sexes regarding the VA area; the females had a wider VA despite having a narrow TF. 

The decreased TF area in the subjects with DTF, with the unchanged VA area, was hypothesized by us as a potential risk for developing VBI following traumas like whiplash in both genders because the VA has less space inside the TF, and it allows for an easier compression of the VA inside the TF. Moreover, a higher VA area in the DTF females may be a different anatomical basis for a higher risk of VBI in females with DTF.

Since TF differs according to gender, age, and side, it is crucial in surgical procedures such as screw fixation in the cervical region and radiological examinations. For this reason, there are studies related to this in the literature. Malik et al. reported significant differences between morphometric parameters in males and females and noted that TF had smaller values in females [20], concurring with our results.

The Zaw et al. [21] study of TF variations in all cervical vertebrae using dry bone in South Africans reported that the prevalence of the normal foramen was higher in females (30.77% in males and 31.54% in females). At the same time, the DTF was higher in males bilaterally (11.54% in males and 6.92% in females). Although it does not seem very accurate to compare this study with our study because it includes all cervical vertebrae, that is, they cannot specify at what level it is, since it is dry bone, we determined that the prevalence of both NTF and DTF (55.9% in NTF; 53.3 in DTF) is high in females [21]. However, Zaw et al. [21] suggested that variations in the FT should be considered during any surgical procedure that involves cervical vertebrae and its related structures. More so, attention should be given to these morphometric differences and variations in sex, age, and laterality during radiological investigations of cervical vertebrae.

We see a similar study by Moreira and Herrero on Brazilians [17] that used CT images from C3 to C7. This study stated that the highest anteroposterior (AP) diameter was seen at the C6 level, and the lateral diameter was at the highest C3 level. The highest value of the area of TF was at the C3 (27.55 mm^2^) level in females and at the C6 level (29.48 mm^2^) in males [17]. Another interesting finding is that all of the variables had higher values on the left side than on the right side, like the findings obtained by Kim et al. [22]. In their study, while the TF gave data on the TF area in normal groups, no finding on the VA area was given. In our study, the fact that there was no side difference in the areas may be related to the samples analyzed.

Regarding the prevalence of DTF, there are studies on different populations ranging from 5.7% to 23.5% in the literature, and most of them are studies using dry bone [23,24]; some of them were made with CT images [25,26]. While some of these studies included all cervical vertebrae, they were made by considering the difference between males and females [25,26]. We also see that some studies were carried out on societies that lived in ancient times [27,28]. The most commonly affected areas are C5 and C6, mostly tending to be located unilaterally. 

In the previous study [29] that was carried out in a similar sample of subjects performed using conventional CT and not CTA, there was an absence of statistically significant differences in the DTF prevalence based on sex. In addition, the author’s research revealed that the presence of DTF was more frequent in C6, followed by C5, C4, and C7. DTF was more prevalent on the right transverse processes, although the difference was not statistically significant. Although Quiles-Guinau et al. [29] only focused on the TF diameters and not on the VA, they found sexual differences in the TF measurements: the lateral diameter, antero-posterior diameter, and the area of TF were bigger in men than in women. Regarding only C6, as we have focused on in the present research, that difference was statistically significant between the sexes for the lateral diameter and the area.

In this study, we showed the effect of a bilateral DTF presence on the TF and VA areas and sexual dimorphism. Most of these are studies evaluating the TF diameters. However, it should be considered when the event is considered as a whole and the idea that the TF/VA area ratio can be used as a differential criterion in evaluating VA diseases [22]. Moreover, the cervical spine is the vertebral region with the most mobility of the complete spinal column. The relevant clinical implications of the TF are obvious in cases of possible compression or trauma of structures that cross it, especially in relation to the VA, depending on whether or not it affects normal blood flow [30], and especially when variations in the TF number and size are involved in the etiology of some clinical syndromes and symptoms like headache, migraine, fainting, VBI as a response to certain neck movements, and blackouts due to low blood pressure in the VA [31]. Thus, the cervical vertebrae and TF morphology are clinically crucial in many ways because of the modifications found in the VA pathway with neurological symptoms such as headache, migraine, fainting, and hearing disorders that may occur due to VA compression. As Zibis et al. [32] stated, knowing the anatomy and variations of TF is imperative, especially in posterior cervical approaches. Approximately 10% of malpractice cases are estimated to be caused by physicians’ lack of knowledge about anatomical variations [33]. 

We have noted an increased VA/TF area ratio in DTF subjects, implying that the VA has less space inside the TF of these subjects. This can be observed in Figure 2 and Figure 3. Moreover, it was previously reported that the VA/TF area ratio would be a valuable tool for diagnosing and evaluating VA disease [22]. 

Previously, different authors proposed that the TF size is associated with the VA size in the manner that higher TF values may be due to higher VA values [34] because the size of the TF is a key determinant of the size of the VA [22], and variations in the size of VAs would affect the morphology of the TF [35]. Nonetheless, other authors concluded that a reduced TF bears no influence on the VA area values [19] and that some anatomic variations, like the DTF, may modify the relationship between the TF and the VA, as we have also found. 

Regarding the increased VA/TF area ratio in the DTF subjects, it is important because it could play a role in favoring a VA transient compression of these subjects under movements like neck rotation or flexion–extension after neck trauma. As the VA enters the TF at C6 and the DTF is most frequently located in C6 [10], knowledge of the VA/TF area ratio in C6 is important in cases of possible compression or trauma of the VA [19].

In this context, the normal range of axial rotation from vertebra C3 to vertebra C7 is, on average, 6° per level, with flexion–extension mean values ranging from 10 to 16° per cervical level [36]. However, the VA3 has been described as being vulnerable to injury because of its anatomical relationships [37]. Also, it is known that secondary vascular compression can be seen due to possible lesions, traumas, general fractures, or luxations in the VA [9]. There is not enough information about anatomical variations like the DTF in relation to the VA that may hypothetically affect the VB blood flow because of VA compression under some circumstances, like whiplash, or that may also hypothetically lead to a VA injury due to the increased VA/TF ratio in DTF subjects when the normal ranges of axial rotation and flexion–extension are increased following neck trauma.

In this context, it has been suggested [19] that the transient mechanical occlusion of the vertebral artery (VA), which can occur in neck trauma such as whiplash or forced head rotation, could further reduce blood flow to the brain and could theoretically be associated with vertebrobasilar transient ischemic attacks (TIAs) in people with double transverse foramen (DTF).

Regarding the importance of the VA/TF area ratio, it is known that unilateral VA compression can be symptomatic [38], although it also has been said that the confluence of bilateral VAs into the basilar artery is the anatomic basis by which unilateral VA occlusion often remains asymptomatic [39]. However, whiplash appears to impact the VA blood flow and causes symptoms of VBI [40]. Also, the DTF subjects present more frequently with known clinical symptoms such as acute headache, dizziness, and vomiting than normal subjects after whiplash [41]. Thus, it is plausible that DTF subjects may be at a higher risk of developing clinical VA-related symptoms than NTF subjects under special circumstances like whiplash or neck traumas. The same may happen in the case of Bow Hunter’s syndrome, a symptomatic VBI caused by the mechanical occlusion of the VA during head rotation [42]. Suppose DTF subjects have less space for the VA inside the TF. In that case, it is also plausible that they may have a higher risk of suffering the related mechanical occlusion of the VA, known as Bow Hunter syndrome, during head rotation. Nevertheless, radiologists, neurologists, and neurosurgeons usually do not search anatomical variations like the DTF in their patients to analyze the possible relationship between their symptoms and their spine anatomy. 

Based on the above, anatomical TF and VA CT studies may be included in the clinical management and follow-up of those subjects presenting VBI symptoms after neck traumas. 

In sum, we showed that bilateral DTF reduced the TF area in a research study that was carried out using a small number of cases. In contrast, the VA area did not change despite the decreasing TF area; it might be an anatomical risk for VBI in subjects with DTF, especially in females, because the VA has less space inside the TF in DTF subjects than in NTF subjects, and that may lead to an easy compression of the VA in DTF subjects following neck trauma. This anatomical research will serve as a starting point for further research to be carried out by clinicians to detect possible anatomical potential VBI risks.

## Figures and Tables

**Figure 1 diagnostics-13-03029-f001:**
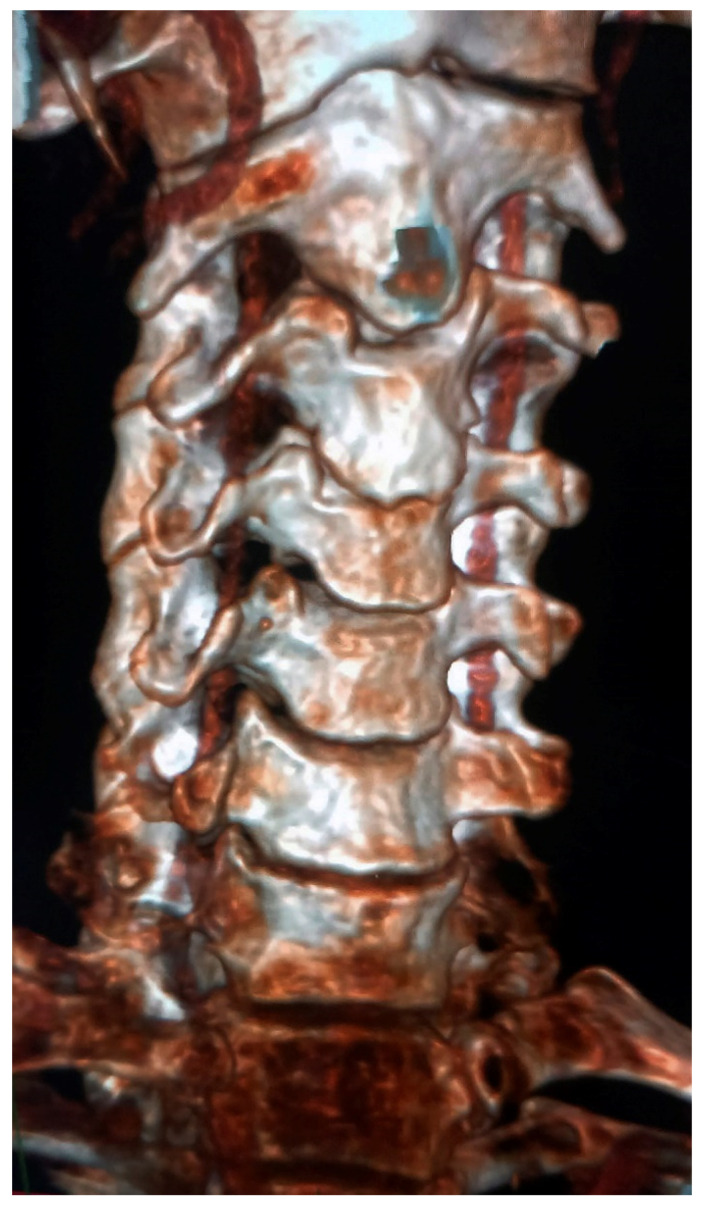
Computed tomography angiography image that shows an anterior view of the cervical spine and associated vertebral arteries.

**Figure 2 diagnostics-13-03029-f002:**
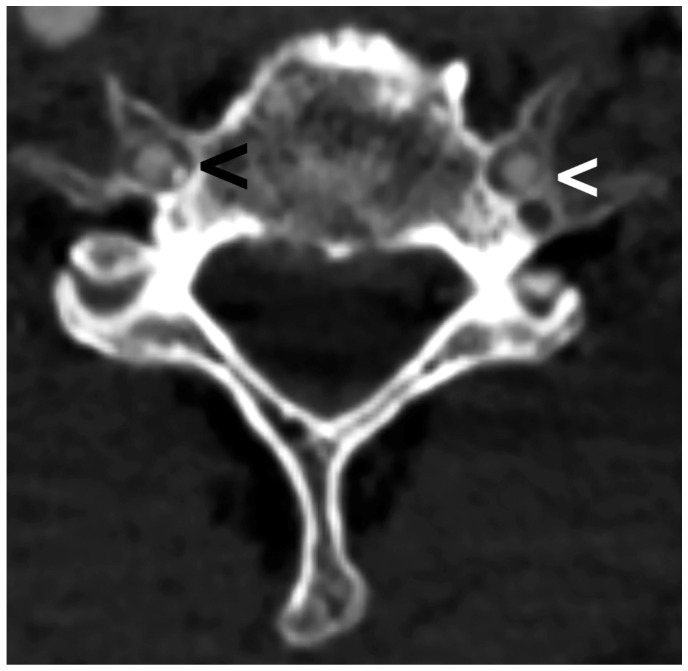
Computed tomography axial view showing the vertebral artery passing through the principal transverse foramina of a vertebra with double transverse foramina (white arrowhead) and through the transverse foramina of a normal transverse foramina vertebra (black arrowhead).

**Figure 3 diagnostics-13-03029-f003:**
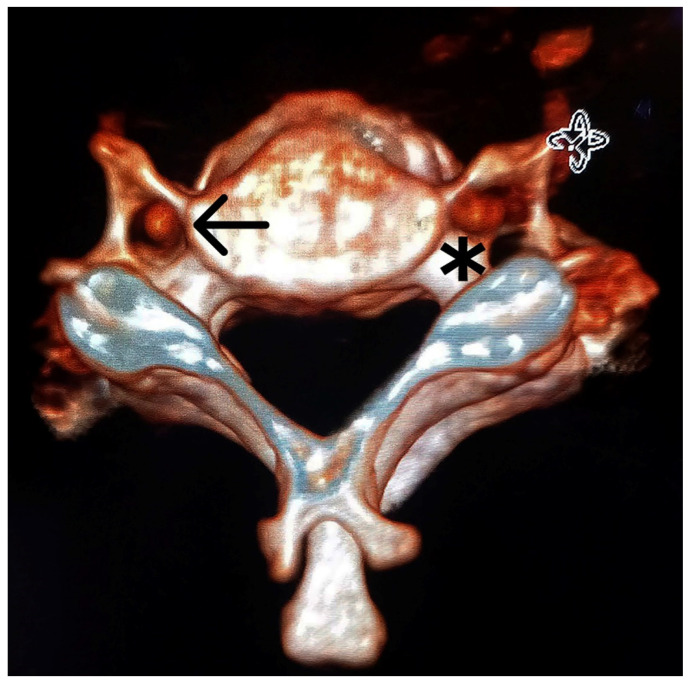
Three-dimensional volume-rendering CT image that shows the transversal view of a vertebra with the vertebral artery (VA) passing through a normal transverse foramen (TF) (indicated by the arrow) and the vertebral artery passing through the principal transverse foramina of a vertebra with a double transverse foramina (indicated by the asterisk).

**Figure 4 diagnostics-13-03029-f004:**
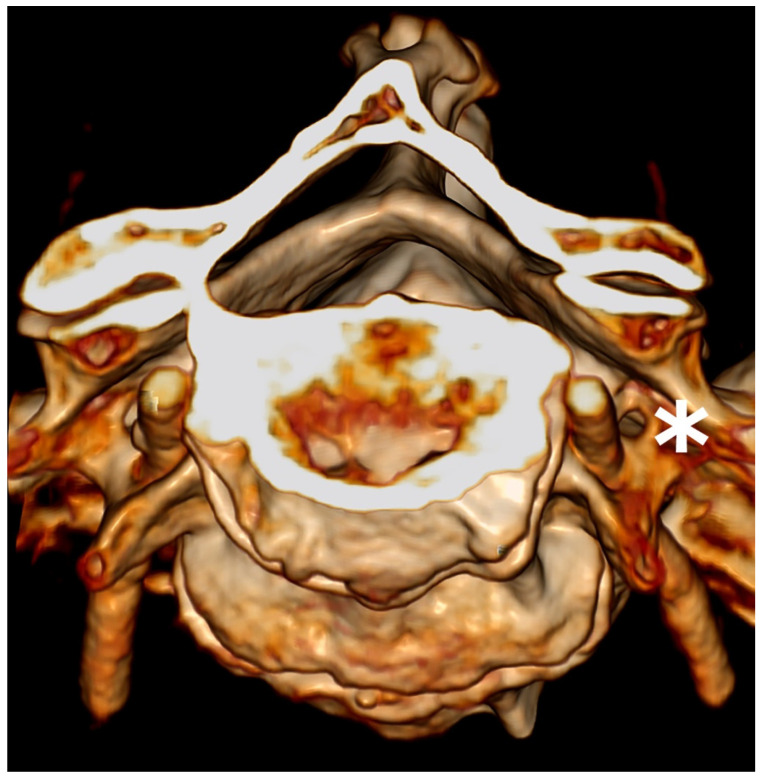
Three-dimensional volume-rendering computed tomography showing the vertebral artery passing through the principal transverse foramina of a vertebra with double transverse foramina (asterisk).

**Table 1 diagnostics-13-03029-t001:** Descriptive values of the variables for the normal transverse foramen group in males and females and side differences (mm^2^).

		Area of the Transverse Foramen			Area of Vertebral Artery		
		Mean ± S.D.	Range	*p*-Value	Mean ± S.D.	Range	*p*-Value
Male (*n* = 26)	Right	30.63 ± 3.95	21.50–48.10		13.55 ± 2.13	10.10–18.10	
	Left	30.00 ± 6.14	22.90–38.10	0.310	13.32 ± 1.58	10.10–16.70	0.670
Female (*n* = 33)	Right	28.08 ± 3.41	20.70–32.40		13.60 ± 2.39	10.70–19.50	
	Left	26.88 ± 2.79	22.30–31.80	0.233	13.71 ± 1.76	10.00–17.40	0.476

S.D.: standard deviation.

**Table 2 diagnostics-13-03029-t002:** Descriptive values of the variables for the double transverse foramen group in males and females and side differences (mm^2^).

		Area of the Transverse Foramen			Area of Vertebral Artery		
		Mean ± S.D.	Range	*p*-Value	Mean ± S.D.	Range	*p*-Value
Male (*n* = 24)	Right	21.20 ± 2.49	15.40–25.80		13.53 ± 1.17	11.90–16.60	
	Left	21.05 ± 2.18	16.60–26.40	0.621	13.53 ± 1.39	11.50–16.50	0.421
Female (*n* = 21)	Right	21.37 ± 1.77	17.70–25.10		14.02 ± 1.21	11.80–16.70	
	Left	21.42 ± 1.90	16.50–25.60	0.924	14.18 ± 0.96	11.90–15.50	0.655

S.D.: standard deviation.

**Table 3 diagnostics-13-03029-t003:** Comparative statistics for males and females in each group (mm^2^).

	Area of the Transverse Foramen			Area of the Vertebral Artery		
	Male	Female	*p*-Value	Male	Female	*p*-Value
NTF	30.31 ± 4.52	27.48 ± 1.69	0.006 *	13.43 ± 1.23	13.65 ± 1.72	0.879
DTF	21.13 ± 2.23	21.39 ± 1.77	0.662	13.67 ± 1.70	14.10 ± 0.91	0.184
*p*-value	0.001 *	0.001 *	---	0.485	0.032 *	---

NTF: normal transverse foramen; DTF: double transverse foramen; * statistically significant.

**Table 4 diagnostics-13-03029-t004:** Ratio vertebral artery area/transverse foramen area.

	Male	Female	*p*-Value
Normal transverse foramen	0.46 ± 0.08	0.51 ± 0.08	0.002 *
Double transverse foramen	0.65 ± 0.08	0.66 ± 0.06	0.556
*p*-value	0.001 *	0.001 *	---

* Statistically significant.

## Data Availability

The datasets generated and/or analyzed during this study are not publicly available, as CT data and DICOM headers contain patient information. Data can be obtained on reasonable request from the corresponding author.

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
