# Peer review of "Double Transverse Foramina—An Anatomical Basis for Possible Vertebrobasilar Insufficiency Risk and Vertebral Artery Injury"

_diagnostics, 2023, doi:10.3390/diagnostics13193029_

Round 1
Reviewer 1 Report
Interesting study:
Only few remarks - discussing double transverse foramen how to distinguish it from retrotransverse foramen (an extensive study has been lastly presented in Journal of Anatomy). I think that this variation should be considered, as well as intro should contain more "philosophy" on normality and anatomical variations as shown in: Translational Research in Anatomy Volume 23, June 2021, 100105 Anatomical normality and variability: Historical perspective and methodological considerations
, , Joe Iwanaga b d f, , , https://doi.org/10.1016/j.tria.2020.100105
Next: I wonder if a better Figure than No 2 can be found - the contours of DTF are poorly visible - and caused my concerns about retrotransverse foramen.
Most of the problems in the vertebrobasilar circle are associated with the vertebral artery impingement - but is it always caused by obliteration, as we see in DTF?
Material and Methods: for descriptive analyses there are some patters of description for metaanalysies, reports, reviews etc. See B. Henry
References: see Translational Research in Anatomy Volume 22, January 2021, 100099 Preliminary study on foramen transversarium of typical cervical vertebrae in KwaZulu-Natal population: Age and gender related changes
, , , , , , https://doi.org/10.1016/j.tria.2020.100099
Author Response
We want to thank the positive reviewer 1 for the valuable comments about our manuscript.
The reviewer suggested minor changes and clarifications that we have carried out and that we detail point by point:
1. Reviewer 1 expressed the possibility of a misdiagnosis/confusion between a DTF and a retrotransverse foramen (RTF) in Figure 2/manuscript on the basis of the recently published article about RTF in J Anat (Pękala JR, Tempski J, Krager E, Johansen J, Lazarz DP, Walocha JA, Tubbs RS, Tomaszewski KA. Systematic review and meta-analysis of the prevalence of the retrotransverse foramen of the atlas. J Anat. 2023 Oct;243(4):570-578. doi:10.1111/joa.13894).
In this context, we believe that our diagnosis of DTF is correct because the RTF is a nonmetric variant “only presented in the atlas vertebra or C1” (PÄ™kala et al., 2023).
As we have analyzed C6 vertebrae and no atlases (C1 vertebra), and the RTF is a variant only presented in the atlas vertebra, we believe there is no possible misdiagnosis between DTF and RTF in our article.
2. We have mentioned in the Introduction section the article by Żytkowski et al (2021) as suggested by the reviewer (please see red colored lines 59-65, reference number 6).
3. We have mentioned in the manuscript the causes of VBI as suggested by the reviewer (please see red colored lines 96-110).
4. We have mentioned the article by Zaw et al (2021) as suggested by the reviewer (please see reference number 21, red colored lines 219-229).
---------------------------------------------------
Reviewer 2 Report
The manuscript has an adequate structure, its sample size and inclusion criteria are correct, as well as its statistics.
Regarding their results, these are reported correctly and clearly. His discussion is pertinent and updated with correct references.
Author Response
We want to thank the positive evaluation of our research manuscript as voiced in the comments by Reviewer 2.
---------------------------------------------------
Reviewer 3 Report
I think this manuscript is well written. And I think it is an interesting piece of research.
The problem may be the small number of cases.
Author Response
We want to thank the positive evaluation of our research by Reviewer 3.
We have commented as suggested by the reviewer on the small number of cases (please, see lines 321 and 322).